# Clinical and Biochemical Profile Associated with Renal Recovery after Acute Kidney Injury in A Mexican Population: Retrospective Cohort Study

**DOI:** 10.3390/medicina59050889

**Published:** 2023-05-06

**Authors:** Josué I. Ruiz-Gallardo, Enrique Cervantes-Pérez, Andrea Pérez de Acha-Chávez, Guillermo A. Cervantes-Cardona, Sol Ramírez-Ochoa, Adriana Nápoles-Echauri, Alejandro González-Ojeda, Clotilde Fuentes-Orozco, Francisco Javier Hernández-Mora, Eduardo Gómez-Sánchez, Jorge I. Michel-González, Carlos Miguel González-Valencia, Gabino Cervantes-Guevara

**Affiliations:** 1Department of Internal Medicine, Hospital Civil de Guadalajara “Fray Antonio Alcalde”, Guadalajara 44350, Mexico; josue.ruizg@icloud.com (J.I.R.-G.); enrique19896@msn.com (E.C.-P.); ramirez_ochoa_sol@hotmail.com (S.R.-O.); meidevilll@gmail.com (J.I.M.-G.); 2Tlajomulco Universitary Center, Universidad de Guadalajara, Tlajomulco de Zúñiga 44100, Mexico; 3Department of Geriatrics, Instituto Nacional de Ciencias Medicas y Nutrición Salvador Zubirán, Mexico City 14080, Mexico; andrea.pdac@gmail.com (A.P.d.A.-C.); izardetheodoroth@gmail.com (A.N.-E.); 4Department of Philosophical, Methodological and Instrumental Disciplines, Health Sciences University Center, Universidad de Guadalajara, Guadalajara 44100, Mexico; gacervantes66@hotmail.com; 5Biomedical Research Unit 02, Hospital de Especialidades, Centro Médico Nacional de Occidente, Guadalajara 44350, Mexico; avygail5@gmail.com (A.G.-O.); clotilde.fuentes@gmail.com (C.F.-O.); 6Department of Human Reproduction, Health Sciences University Center, Universidad de Guadalajara, Guadalajara 44329, Mexico; frank.gine@gmail.com; 7Division of Clinical Disciplines, Health Sciences University Center, Universidad de Guadalajara, Guadalajara 44100, Mexico; eduardo.gomez@cucs.udg.mx; 8Department of Research Ethics, Hospital Hispano, Guadalajara 44140, Mexico; carlosm.gvalencia@gmail.com; 9Department of Welfare and Sustainable Development, Centro Universitario del Norte, Universidad de Guadalajara, Colotlán 46200, Mexico; 10Department of Gastroenterology, Hospital Civil de Guadalajara “Fray Antonio Alcalde”, Guadalajara 44350, Mexico

**Keywords:** acute kidney injury, AKI, AKI remission, chronic kidney disease

## Abstract

*Background and Objectives*: Our primary objective was to study the clinical and biochemical characteristics associated with acute kidney injury (AKI) remission in a group of Mexican patients. *Materials and methods*: We retrospectively enrolled 75 patients who were diagnosed with AKI and separated the sample into two groups: nonremitting patients (*n* = 27, 36%) vs. remitting patients (*n* = 48, 64%). *Results*: We found significant relationships between nonremitting AKI and previous diagnosis of chronic kidney disease (*p* = 0.009), higher serum creatinine (Cr) at admission (*p* < 0.0001), lower estimated glomerular filtration rate (eGFR) (*p* < 0.0001), maximum serum creatinine during hospitalization (*p* < 0.0001), higher fractional excretion of sodium (FENa) (*p* < 0.0003) and 24-h urine protein (*p* = 0.005), higher serum potassium on admission (*p* = 0.025), abnormal levels of procalcitonin (*p* = 0.006), and increased risk of death (*p* = 0.015). *Conclusion*: Chronic kidney disease (CKD), lower eGFR, higher levels of serum creatinine during hospitalization, higher FENa and 24-h urine protein, abnormal levels of procalcitonin, and higher serum potassium on admission were associated with nonremitting AKI. These findings may facilitate the rapid identification of patients at risk for nonremitting AKI based on clinical and biochemical characteristics. Furthermore, these findings may inform the design of timely strategies for the vigilance, prevention, and treatment of AKI.

## 1. Introduction

Acute kidney injury (AKI) refers to an abrupt diminution of renal function that leads to retention of urea, nitrogenous waste products, and dysregulation of liquid volume and electrolytes [1].

The Kidney Disease Improving Global Outcomes (KDIGO) guidelines define AKI as an increase in serum creatinine above or equal to 0.3 mg/dl in 48 h, an increase above or equal to 1.5 times the basal level in the last 7 days, or a diminution of urinary output (<0.5 mL/kg/h) for at least six hours. It is classified by stages (1 to 3) according to the degree of increment in serum creatinine level or decrease in urinary output [1].

AKI is associated with diminished survival 10 years after the episode, even in mild cases [2]. The reported global incidence for AKI is estimated to be 5000 cases per 1,000,000 persons a year [3]. The estimated global prevalence of AKI acquired in the hospital is 21% [4].

A single event of AKI, even at lower stages and with total resolution of the injury, can lead to a poorer functional prognosis and survival in comparison with patients who do not develop AKI [5]. There are various clinical characteristics associated with susceptibility to developing AKI, including being over 60 years old, diagnosed with type 2 diabetes mellitus (T2DM), chronic renal disease (CRD), cardiovascular disease, or cancer, and being a user of drugs that intervene in the microrhenal circulation, among others [6]. The most well-researched, strongly associated, and most commonly used biomarker of AKI is the elevation of serum creatinine levels [7] because of its cost-benefit ratio and prognostic capacity to demonstrate that there is a proportional elevation of morbidity and mortality risk associated with the elevation of serum creatinine levels [7]. Other biochemical markers used more recently (cystatine-C, neutrophil gelatinase-associated lipocalin (NGAL), tissue inhibitor of metalloproteinases 2 (TIMP2), insulin-like growth factor binding protein 7 (IGFBP7), and kidney injury molecule 1 (KIM1), to mention a few [8,9], are difficult to assess in certain hospital settings [9]. Therefore, extensive research into other factors associated with nonremitting AKI is needed.

There is little evidence for the relationship between previously-described clinical and biochemical markers and the severity and resolution of AKI, especially in the Mexican population [10]. We aimed to identify an association between clinical and biochemical characteristics and AKI remission in a population of Mexican patients.

## 2. Materials and Methods

We retrospectively enrolled patients admitted to the Unit of Internal Medicine of the Hospital Civil de Guadalajara Fray Antonio Alcalde (HCG) between March 2019 and January 2020 who were diagnosed with AKI in accordance with the Kidney Disease Improving Global Outcomes (KDIGO) [1] guidelines at any time during their hospital stay. Eligible patients were required to have all of the variables of interest in their medical records and a blood and urine sample analysis performed after the AKI diagnosis. Patient demographics, clinical and laboratory data, and outcomes were extracted from patient medical records.

The primary objective of this study was to test the association between clinical and biochemical characteristics and AKI remission in this sample of patients. The study was conducted in accordance with the Declaration of Helsinki. The Clinical Research and Bioethics Committee of the Hospital Civil de Guadalajara Fray Antonio Alcalde approved this study (R-169/22), and the requirement for informed consent for this retrospective analysis was waived.

### Data and Statistical Analysis

Sample size was calculated with Epi-Info software (CDC, Atlanta, GA, USA). We used data reported by Panichote et al. [11] as a reference for the expected rate of AKI remission. The sample size required for a 95% confidence interval (CI) with a power of 80% was 10 patients.

Patients with AKI remission were defined as those who did not meet the AKI definition in accordance with the KDIGO guidelines before their hospital discharge [1]. For the analysis of the data in reference to the evaluation of AKI, only the first AKI event during hospitalization was considered, diagnosed according to the KDIGO criteria [1]. For the initial selection of patients (Figure 1), the maximum creatinine levels were considered during this initial AKI event and until the resolution of the event. Therefore, reassessments or multiple episodes of AKI between the first event and hospital discharge were no longer considered.

The normality and distribution of the quantitative variables of interest were established with the Kolmogorov-Smirnov test [12] using a parametric analysis. For the descriptive analysis, parametric quantitative variables were analyzed with measures of central tendency and dispersion, mean and standard deviation (SD), and non-parametric quantitative variables with median and interquartile range (IQR).

For qualitative variables, proportions were used. For inferential statistics, we used the chi-square test to compare the proportions of categorical dichotomous variables. For the comparison of independent means, we used the Student’s *t* test. All *p* values < 0.05 were considered significant. For the comparison of independent medians of variables with nonnormal distributions, the Mann-Whitney test was used, and 95% confidence intervals (95% CIs) were used. All data were processed with SPSS software, version 23.0 (IBM, Armonk, NY, USA).

Multivariate analysis was carried out using an unconditional logistic regression model expressed as an odds ratio (OR). To test the independence of the risk factors, the variables considered significant in the univariate analysis (*p* < 0.05) were entered into a multivariate logistic regression model with likelihood ratio backward selection and a significance criterion of *p* < 0.05.

## 3. Results

We analyzed the data for 75 patients who met the inclusion criteria (Figure 1). Female patients were more prevalent in the sample (*n* = 39, 52%). The most frequent diagnoses on admission were cardiorenal syndrome (CRS) and pneumonia (*n* = 13, 17.3% for each of these diagnoses). The mean hospitalization stay was 10 days (SD = 7.8). There was a significant difference (*p* = 0.015) in the deaths reported for each group: six (22.2%) for patients with remitted AKI versus two (4.17%) for patients with nonremittent AKI.

Table 1 shows the clinical characteristics of the two groups of patients: remitting vs. nonremitting.

Table 2 shows the underlying diagnosis at admission of the patients included in this study.

Table 3 shows the differences in biochemical characteristics between patients with remitting vs. nonremitting AKI.

For the multivariate analysis, we used a binomial logistic regression with a backward step model of entry for the seven significant variables of the univariate analysis: chronic kidney disease, days of hospital stay, serum Cr, maximum serum Cr, FENa, serum K+, and procalcitonin. Of these, an independent risk factor for nonremitting AKI was identified: maximum serum Cr [Risk Ratio = 10.85 (2.73–66.99); *p* = 0.0031]. Table 4 shows the results for all the variables analyzed in the multivariate analysis.

## 4. Discussion

Here, we identified a clear association between multiple clinical and biochemical variables and remission of AKI. For statistical analysis of the data, we separated the sample of patients into two groups: nonremitting (*n* = 27, 36%) vs. remitting (*n* = 48, 64%). We did not find significant differences between groups in the clinical characteristics of the patients included in the sample, except for chronic kidney disease (CKD) antecedent to AKI presentation. In contrast, prior findings [13], for example, a higher Charlson Comorbidity Index score, have been reported as a risk factor for the development of CKD after AKI [14]. Other associated comorbidities are congestive heart failure, decompensated liver disease, acute coronary syndrome, and malignancy, which are risk factors for recurrent AKI after the index AKI episode [15]. Our findings may differ from those of prior studies because the population we analyzed was more homogenous in sociodemographic characteristics.

Even when a significant difference is observed between the days of hospitalization between both groups (the duration being longer in the group of patients who did not remit), this finding must be interpreted with care since this may be due to the same non-recovery of the AKI and, therefore, it is not a causal factor. Unfortunately, in the data collected we do not have the information regarding the length of hospitalization prior to the appearance of the AKI episode to make a better causal association.

We observed statistically significant differences in the levels of serum creatinine at admission between remitting and nonremitting patients (Table 3). Similarly, the estimated glomerular filtration rate (eGFR) at admission for remitting vs. nonremitting patients was significantly different (Table 3). In accordance with the findings of Acosta-Ochoa [16], we presume that this finding is correlated with the severity of kidney injury at admission, leading to a decreased probability of recovery. Additionally, in this study, the minimum eGFR, maximum serum creatine, protein urinary excretion in 24 h, and potassium levels were also significantly different between patients with remitting vs. nonremitting AKI [16].

There was also a significant difference in fractional excretion of sodium (FENa) values between patient groups (Table 3). Vanmassenhove et al. [17] associated values of FENa greater than 1% with intrinsic causes of AKI. At the same time, FENa values of less than 1% were associated with transitory causes of AKI and are assumed to correspond with a greater likelihood of AKI remission. To our knowledge, there have been no other studies reporting these findings, and we assume that this lack of corroboration is because of the limitations related to the assumption of preserved tubular function [18]. In addition to this, we made a comparison between groups taking the FENa value of less than 1% as the cut-off point and thus separate between patients with AKI of probable prerenal origin, observing that, in the group of patients who presented remission, there was a higher proportion of patients with FENa < 1% (*n* = 27, 57%) compared to those who did not remit (*n* = 7, 25.93%), with a prevalence ratio of 0.42 (0.20–0.83) and a significant difference between groups (*p* = 0.011). This finding, accompanied by the fact that the remitting group presented a higher proportion of hypovolemia (even though the difference was not significant, Table 1) at the time of AKI diagnosis, may indicate that FENa < 1% may be a better predictor of remission as indicative of prerenal causes of AKI, the treatment of which includes aggressive volume replacement [17,18]. This finding, due to the methodological scope of this study, is only applicable to this cohort, but it provides us with a starting point for future confirmatory studies in the Mexican or international population.

CKD is widely associated with the development, nonremission, morbidity, and mortality of AKI, suggesting changes in tissue and cellular signaling, transforming growth factor beta, hypoxia inducible factor, and other markers of chronic inflammation [19]. Accordingly, we found a statistically significant association between CKD and nonremitting AKI (Table 1).

One important finding of this study was the significant association between the use of angiotensin receptor blockers (ARBs) and nonremitting AKI (*p* = 0.012, OR 5.4, 95% CI 2.01–22.7). The evidence supporting the use or discontinuation of ARB in patients at risk of developing AKI is controversial. The KDIGO guidelines for Chronic Renal Disease suggest discontinuation [20]. However, a recent review by Tomson et al. [21] suggested that the pleiotropic effects of blocking the renin–angiotensin–aldosterone axis surpass the risk associated with AKI. In the context of our study, we believe that the higher risk associated with nonremitting AKI could be secondary to an alteration in the microvessel circulation within the kidney that can preserve hypoperfusion and an inflammatory state.

Another association of interest was related to acute phase reactants (APR). We found a significant association between nonremitting AKI and altered values of serum procalcitonin but not C-reactive protein (CRP) (Table 3). The association with procalcitonin was previously described by Chavez-Iñiguez et al. [22] in the context of sepsis and multiple organic failure. Even though procalcitonin is a marker of inflammation used for the diagnosis, or as a guide to the treatment, of infectious diseases [23], there is an elevation in some groups of patients regardless of the presence of an active infection, this has been reported in patients with CKD [24,25], so the difference presented in our patients may be due to the fact that the patients who did not remit had a significantly higher proportion of CKD prior to the diagnosis of AKI. There is insufficient evidence supporting procalcitonin as a specific marker for kidney injury prognosis; however, there have been some reports that CKD patients without a history of dialysis or infection have significantly higher levels of PCT than healthy individuals—this could be due to a possible increase in pro-inflammatory metabolites in serum caused by decreased renal function, which, in turn, stimulates the immune system and causes the release of PCT into the systemic circulation [25]. Some new biomarkers have recently been investigated in AKI, some of which, such as neutrophil gelatinase-associated lipocalin (NGAL), show promising potential as predictors of the persistence of AKI and the normalization of creatinine levels in the first 48 h [26,27,28].

Patients who did not show AKI remittance had an increased risk of mortality (*p* = 0.015); this association has been previously reported [29] and suggests a persistent inflammatory process independent of the resolution of the other systemic injuries. This study is in accordance with evidence published globally examining the multiple variables associated with nonremitting AKI that can also be applied to the Mexican population. This study also contributed to the evidence for the association of FENa and procalcitonin with nonremitting AKI and underscores the need for future research into these associations. The early detection of AKI may be difficult, and the current diagnostic criteria are dependent on serum creatinine and urine output [30]. Detailed risk stratification and frequent creatinine and urine output monitoring in high-risk populations could enable the earlier diagnosis of AKI. History taking, physical examination, and hemodynamic monitoring in populations with a higher risk of AKI are critical [30].

The objective of the multivariate analysis was to obtain a better predictive ability between the correlated variables, finding that the only independent variable that came out statistically significant was the maximum serum Cr, with a higher maximum serum Cr during the AKI episode, the patients have 10.85 times more chances of non-remission of AKI; logically, the more kidney damage, from a functional point of view, even temporarily, less likely recovery is [18].

Some of the new directions to consider when talking about renal recovery could help us to better stratify AKI [31]. For example, using the trajectory of renal dysfunction in terms of renal events could help us assess the response of a patient to therapeutic interventions and this, in turn, could help us make better subsequent therapeutic decisions [31].

It has previously been reported that AKI recovery patterns aid in stratifying patients with poor outcomes [32,33]. Taking this into account, the ASSESS-AKI cohort [31] focused on outcomes based on the renal events of patients who did not achieve AKI recovery. A 51% increased risk of major renal events in patients who did not recover compared to those who recovered was associated with an increased incidence of CKD in the group.

A history of CKD has been identified as being associated with impaired renal recovery [31]. In the ASSES-AKI cohort [31], having CKD prior to the AKI event was the greatest differentiating component between patients who presented recovery and those who did not. However, the increased risk of major renal events remains even after removing this factor [31]. This could be due to some other event that was not considered in the follow-up, such as the number of AKI events or the use of nephrotoxic drugs [31].

As seen when comparing the different results reported in the scientific literature, the ability to distinguish recovery from nonrecovery, and therefore the recovery time, depends on the definition used to measure recovery [34]. Xu et al. [35] compared the five most common definitions of renal recovery and found frequencies of recovery ranging from 44% to 84%, showing the need for unified definitions to address this issue. This highlights the need for future research to determine the variables associated with susceptibility, necessary care, and outcomes [36]. The availability of big data and thier analysis with new computational tools, as well as new biomarkers, could offer opportunities to further understand the interactions related to renal recovery [37,38].

## 5. Conclusions

Nonremitting AKI is associated with CKD, lower eGFR, higher levels of serum creatinine during hospitalization, higher FENa and 24-h urine protein, abnormal levels of procalcitonin, and higher serum potassium on admission. These data could facilitate the rapid identification of patients with clinical and biochemical characteristics associated with nonremission of AKI, especially maximum serum creatinine, which at a higher value showed 10.85 times more probability of a nonremitting acute kidney injury. These findings may inform the design of timely strategies for the vigilance, prevention, and treatment of AKI. More research is needed to develop better interventions for at-risk patients to prevent AKI, as this is a relevant public health problem. Elucidating the factors associated with their pathophysiology, guidelines for effective therapeutic interventions reduce the possibility of permanent kidney damage or severe adverse events, leading to a better understanding of AKI.

## Figures and Tables

**Figure 1 medicina-59-00889-f001:**
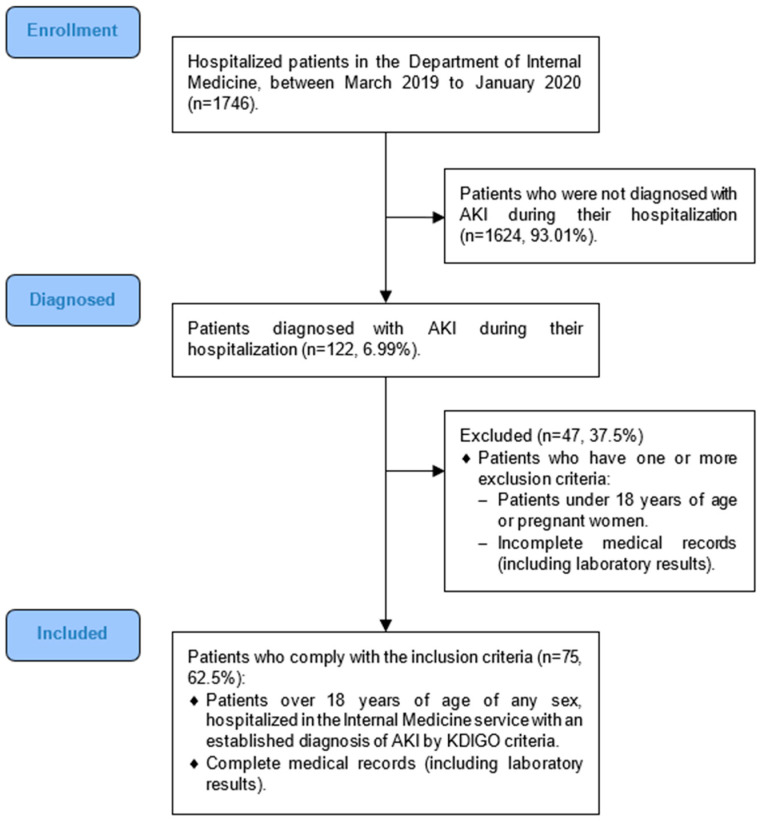
Recruitment and selection of patients.

**Table 1 medicina-59-00889-t001:** Clinical and sociodemographic characteristics of the population studied.

Variable	Non-Remitting (*n* = 27)	Remitting (*n* = 48)	PR (IC 95%)	*p*
Age (years), mean (SD)	49.26 (17.64)	50.94 (14.95)	-	0.663
Masculine sex, n (%)	16 (59.26)	20 (41.67)	1.57 (0.86–2.94)	0.143
BMI (kg/m^2^), mean (SD)	31.63 (6.69)	32.98 (6.77)	-	0.408
Diabetes, n (%)	11 (40.74)	30 (62.50)	0.57 (0.30–1.04)	0.069
HBP, n (%)	17 (62.96)	24 (50.00)	1.41 (0.76–2.69)	0.279
Smoking, n (%)	6 (22.22)	9 (18.75)	1.14 (0.52–2.13)	0.718
COPD, n (%)	1 (3.70)	4 (8.33)	0.53 (0.09–1.81)	0.440
Cardiopathy, n (%)	1 (3.70)	2 (4.17)	0.92 (0.16–2.44)	0.921
Stroke, n (%)	1 (3.70)	1 (2.08)	1.40 (0.26–2.95)	0.675
Chronic kidney disease, n (%)	13 (48.14)	10 (20.83)	2.09 (1.16–3.69)	0.013
ARB use, n (%)	9 (33.33)	9 (18.75)	1.58 (0.83–2.77)	0.155
Hypovolemia, n (%)	3 (11.11)	11 (22.92)	0.54 (0.18–1.31)	0.207
Vasopressor use, n (%)	7 (25.93)	6 (12.50)	1.66 (0.83–2.90)	0.140
Shock, n (%)	7 (25.93)	6 (12.50)	1.66 (0.83–2.90)	0.140
Multiorgan failure, n (%)	8 (29.63)	8 (16.67)	1.55 (0.79–2.72)	0.188
Cardiorenal syndrome, n (%)	4 (14.81)	9 (18.75)	0.82 (0.32–1.73)	0.665
Pneumonia, n (%)	3 (11.11)	11 (22.92)	0.54 (0.18–1.31)	0.207
Septic shock, n (%)	3 (11.11)	3 (6.25)	1.43 (0.51–2.72)	0.456
Urinary tract infection, n (%)	6 (22.22)	4 (8.33)	1.85 (0.90–3.16)	0.089
Diabetic ketoacidosis, n (%)	1 (3.70)	6 (12.50)	0.37 (0.06–1.42)	0.208
AKI 1, n (%)	9 (33.33)	24 (50)	0.63 (0.32–1.18)	0.162
AKI 2, n (%)	4 (14.81)	17 (35.42)	0.44 (0.17–1.01)	0.056
AKI 3, n (%)	14 (51.85)	7 (14.58)	2.76 (1.56–4.85)	0.0006
Days of hospital stay, median (IQR)	11 (11)	6 (5)	-	0.0005

SD = standard deviation, BMI = body mass index, HBP = high blood pressure, COPD = chronic obstructive pulmonary disease, ARB = angiotensin receptor blockers, RR= risk ratio, IC = confidence interval, AKI = acute kidney injury, IQR = interquartile range.

**Table 2 medicina-59-00889-t002:** Underlying diagnosis at admission of the population studied.

Non-Remitting AKI	
Underlying diagnosis at admission	*n* (total = 27)
Dress syndrome	1 (3.7%)
Epilepsy	1 (3.7%)
Ischemic stroke	1 (3.7%)
Cardiorenal syndrome	4 (14.81%)
Anemic syndrome	1 (3.7%)
Pneumonia	3 (11.11%)
Urinary tract infection	6 (22.22%)
Hemorrhagic stroke	1 (3.7%)
Infectious gastroenteritis	2 (7.41%)
Septic shock	3 (3.7%)
Abdominal sepsis	1 (3.7%)
Diabetic ketoacidosis	1 (3.7%)
Ethylic suppression	1 (3.7%)
Dengue	1 (3.7%)
Remitting AKI	
Underlying diagnosis at admission	*n* (total = 48)
Pneumonia	11 (22.92%)
Urinary tract infection	4 (8.33%)
Pancreatitis	1 (2.08%)
Diabetic ketoacidosis	6 (12.5%)
Hypothyroidism	1 (2.08%)
Cardiorenal syndrome	9 (19.75%)
Systemic lupus erythematosus	1 (2.08%)
Congestive heart failure	1 (2.08%)
Sjogren syndrome	1 (2.08%)
Non Hodgkin lymphoma	1 (2.08%)
Hypertensive crisis	1 (2.08%)
Ethylic suppression	3 (6.25%)
Sepsis	3 (6.25%)
Meningitis	1 (2.08%)
Ischemic stroke	1 (2.08%)
Dengue	2 (4.17%)
Diabetic gastroparesis	1 (2.08%)

**Table 3 medicina-59-00889-t003:** Biochemical characteristics of the two groups of patients analyzed.

Variable	Non-Remitting, *n* = 27, Median, (IQR)	Remitting, *n* = 48, Median, (IQR)	*p*
Serum Cr (mmol/L)	223.70 (155.6)	126.40 (61.94)	<0.0001
eGFR (ml/min/1.73 m^2^)	22 (15)	45 (35.37)	<0.0001
Maximum serum Cr (mmol/L)	358.1 (287.4)	136.6 (85.1)	<0.0001
Minimum eGFR (ml/min/1.73 m^2^)	14(16.1)	42 (33.77)	<0.0001
Protein urinary excretion in 24 h (mg/day)	1400 (2200)	475 (975.5)	0.005
FENa (%)	1.60 (1.22)	0.80 (0.87)	<0.0003
Urine Na (mmol/day)	74 (51)	54.50 (68.75)	0.585
Urinary Cr (mmol/L)	6.48 (2.64)	7.16 (2.62)	0.756
Serum Na (mmol/L)	137 (5)	135.5 (5)	0.419
Serum K^+^ (mmol/L)	4.10 (1.3)	3.60 (0.75)	0.025
Uric acid (mmol/L)	416.40 (148.7)	416.40 (173.9)	0.596
Procalcitonin (ng/mL)	0.70 (2.3)	0.34 (0.5)	0.006
C-reactive protein (mg/L)	50 (54)	41 (62.5)	0.786

SD = standard deviation, Cr = creatinine, eGFR = estimated glomerular filtration rate, FENa = fractional excretion of sodium, IQR = interquartile range.

**Table 4 medicina-59-00889-t004:** Logistic regression model for variables associated to non-remitting AKI.

Variable	OR (IC 95%)	*p*
Chronic kidney disease	0.23 (0.01–1.60)	0.182
Days of hospital stay	1.01 (0.87–1.14)	0.833
Serum Cr	0.57 (0.12–2.25)	0.433
Maximum serum Cr	10.85 (2.73–66.99)	<0.0031
FENa	1.28 (0.45–3.60)	0.6283
Serum K^+^	1.34 (0.49–3.76)	0.557
Procalcitonin	0.99 (0.92–1.04)	0.778

Cr = creatinine, eGFR = estimated glomerular filtration rate, FENa = fractional excretion of sodium, OR = odds ratio, IC = confidence interval.

## Data Availability

The data presented in this study are available on request from the corresponding author. The data are not publicly available due to privacy reasons.

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
