# Peer review of "Clinical and Biochemical Profile Associated with Renal Recovery after Acute Kidney Injury in A Mexican Population: Retrospective Cohort Study"

_medicina, 2023, doi:10.3390/medicina59050889_

Round 1

Reviewer 1 Report

The study aimed to investigate the clinical and biochemical characteristics associated with AKI remission in a group of Mexican patients. They found that CKD, lower eGFR, higher levels of serum creatinine during hospitalization, higher FENa and 24-hour urine protein, abnormal levels of procalcitonin, and higher serum potassium on admission were associated with nonremitting AKI. While the study provided valuable insights into the clinical and biochemical factors associated with nonremitting AKI in the Mexican population, overall, it was properly organized. However, the procedure is not clear in the methods and some concerns listed below limit the clear narrative of the current study.

Comments:

1. Were the patients' underlying etiology and common causes of AKI considered at the time of AKI diagnosis? Different etiologies and causes of AKI can lead to varying degrees of recovery time and prognosis, and understanding these factors may further elucidate the associations between clinical and biochemical profiles and AKI remission.

2. Subgroup analysis based on AKI staging is suggested because the degree of AKI severity largely affects recovery time and outcomes.

3. Subgroup analysis based on the presence of underlying CKD is also suggested since patients with CKD are well-known to be at a higher risk of developing nonremitting AKI.

4. How were the data selected from patients who underwent multiple AKI evaluations during hospitalization? It is important to know how the authors selected which data points to include in the analysis and how this could have influenced the results.

5. Since the study only observed AKI cases that occurred within the hospital setting, what about the duration of hospital stay or the time from AKI onset to observation endpoint? Could these factors have affected the results?

Author Response

Comments:

  1. Were the patients' underlying etiology and common causes of AKI considered at the time of AKI diagnosis? Different etiologies and causes of AKI can lead to varying degrees of recovery time and prognosis, and understanding these factors may further elucidate the associations between clinical and biochemical profiles and AKI remission.

To clarify this point, we included in Table 1 some comparisons of underlying diagnoses for both groups of patients (only those who had at least one event in both groups were compared) and in the manuscript, the frequencies (percentage) and number of patients for each etiology of admission for each group of patients.

  1. Subgroup analysis based on AKI staging is suggested because the degree of AKI severity largely affects recovery time and outcomes.

We include in Table 1 the comparison by AKI stage between each group of patients.

  1. Subgroup analysis based on the presence of underlying CKD is also suggested since patients with CKD are well-known to be at a higher risk of developing nonremitting AKI.

Table 1 already describes the comparison between groups on the history of underlying CKD, reporting a significant difference (p=0.009) with 48.14% of patients with CKD prior to the diagnosis of AKI in patients who did not present remission compared with only 20.83% in those who presented remission.

  1. How were the data selected from patients who underwent multiple AKI evaluations during hospitalization? It is important to know how the authors selected which data points to include in the analysis and how this could have influenced the results.

For the analysis of the data in reference to the evaluation of AKI, only the first AKI event during hospitalization was considered, diagnosed according to the KDIGO criteria. For the initial selection of patients, the maximum creatinine levels were considered during this initial AKI event and until the resolution of the event. Therefore, reassessments or multiple episodes of AKI between the first event and hospital discharge were no longer considered. Recruitment process diagram added for clarity. 

  1. Since the study only observed AKI cases that occurred within the hospital setting, what about the duration of hospital stay or the time from AKI onset to observation endpoint? Could these factors have affected the results?

The comparison between the length of hospitalization between both groups was added to Table 1 and interpretation about the result in discussion section.

Reviewer 2 Report

The authors retrospectively examined the factors affecting AKI recovery in single centers in Mexico. his article

Regarding the risk factors related for non- recovery after AKI, it has already been reported that patients with CKD comorbidity, high Cr, and low eGFR have been reported.

There are certainly no reports that FENa is a risk factor for non-recovery AKI, but is this a characteristic of the Mexican population or your cohort?

Judging from the rate of oral ARB administration and the presence or absence of dehydration, is it possible that prerenal AKI due to dehydration or hypotension was more common in the recovery group?

In general, it is believed that prerenal AKI is easier to recover than renal AKI such as ATN with appropriate infusion and blood pressure control.

Could you discuss why FENa showed a significant difference between the two groups in this study, but not in other studies?

Did urinary protein show a normal distribution?

If not, you should show the median value instead of average.

Did FENa also show a normal distribution? If not there's no point in comparing the average values (1.76 vs 1.03). It is better to compare the rate(%) of less than 1% that is considered prerenal.

Forni et. Al. described that risk factors for non-recovery of AKI are age, CKD, comorbidity, higher severity of AKI and acute disease scores. Second, AKI and CKD are mutually related and seem to have a common denominator. You cited as References 5)

You should consider what caused this discrepancy between procalcitonin and CRP. It is important.

Could you perform Multivariate analysis to determine which variables would be significant risk factors for non-recovery AKI.

Author Response

There are certainly no reports that FENa is a risk factor for non-recovery AKI, but is this a characteristic of the Mexican population or your cohort?

At this moment and according to the scope and methodology used to calculate the sample size, we could infer that this characteristic applies to the analyzed cohort; however, it can serve as a starting point to carry out a study with a larger population scope. This point is clarified in the discussion.

Judging from the rate of oral ARB administration and the presence or absence of dehydration, is it possible that prerenal AKI due to dehydration or hypotension was more common in the recovery group?

The comparison using a contingency table of the proportion of patients with FENa less than 1% in each group was added to the results in the text in order to describe patients with prerenal AKI.

Could you discuss why FENa showed a significant difference between the two groups in this study, but not in other studies?

We add to the discussion what we consider a possible explanation of this, also, we mentioned previously in the discussion that one of the possible causes of this can be related to the assumption of preserved tubular function.

Did urinary protein show a normal distribution?

If not, you should show the median value instead of average.

Urinary proteins showed a non-normal distribution, as well as several of the variables analyzed, the comparison and p values (which were reported correctly) are the result of the analysis with the Mann Whiteney U test, the results were adjusted in Table 1 and 2 (table 2 now renamed table 3), in all cases, we report the median accompanied by minimums and maximums

Did FENa also show a normal distribution? If not there's no point in comparing the average values (1.76 vs 1.03). It is better to compare the rate(%) of less than 1% that is considered prerenal.

The comparison between groups was added using a FENa lower than 1% as a cut-off point, as well as what was reported in the general comparison of FENa values was corrected and the median was placed with minimum and maximum values due to its non-normal distribution.

Forni et. Al. described that risk factors for non-recovery of AKI are age, CKD, comorbidity, higher severity of AKI and acute disease scores. Second, AKI and CKD are mutually related and seem to have a common denominator. You cited as References 5)

You should consider what caused this discrepancy between procalcitonin and CRP. It is important.

We address this point in the discussion.

Could you perform Multivariate analysis to determine which variables would be significant risk factors for non-recovery AKI.

We realized the multivariate analysis suggested, we put the procedure in methods and the corresponding results and conclusion in the according sections.

Reviewer 3 Report

Overall, this is a straightforward and interesting retrospective cohort study characterizing the clinical biochemical parameters between Mexican patients with AKI remission vs non remission. The manuscript is very well written, clearly organized and thorough, and the approach is straightforward and clearly described. Conclusions are supported by results data. Therefore, I only have a couple of minor suggestions to improve the manuscript.

1. A consort flow diagram would provide a straightforward idea about the cohort study design, where specifies the inclusion and exclusion criteria.

2. In table 1 and 2, the column variables “Non regression of AKI” and “AKI regression” are not commonly used terms. I suggest authors just use “non-remitting” vs “remitting”, which matches the context.

Author Response

  1. A consort flow diagram would provide a straightforward idea about the cohort study design, where specifies the inclusion and exclusion criteria.

The suggested diagram is added.

  1. In table 1 and 2, the column variables “Non regression of AKI” and “AKI regression” are not commonly used terms. I suggest authors just use “non-remitting” vs “remitting”, which matches the context.

The terms in both tables were changed according to the rest of the text.

Round 2

Reviewer 1 Report

In summary, the results described in this manuscript are not groundbreaking and rather expected. However, they do provide insight into the clinical and biochemical factors linked to renal recovery among AKI patients in the Mexican population. 

Author Response

Thanks for the feedback.

Reviewer 2 Report

The authors retrospectively examined the factors affecting AKI recovery in single centers in Mexico. At first, you described higher Cr level, lower eGFR, maximum serum Cr, higher FENa

24 UP, higher potassium, and proclcitonin (PCT) showed significant in univariate analysis.

But in multivariate analysis, only minimum eGFR (maximum Cr?) showed independent risk factor for non-remitting AKI.

So all of the other factors (higher Cr level, lower eGFR, maximum serum Cr, higher FENa

24 UP, higher potassium, and proclcitonin) may be confounding, which leads to slightly different conclusions.

Show the details of the multivariate analysis in a table. How many variables did you use in multivariate analysis?  Since serum Cr and eGFR are almost synonymous with each other, it is better to use only one of them as a variable to reduce the number of variable in multivariate analysis.

Data that do not show a normal distribution have been corrected from average to median. Let's also modify (minimum-max) to (IQR).

FENAFENa  (in Discussion   L194)

In Discussion

Ref 25 mentioned that Procalcitonin is elevated in inflammatory bowel disease such as CD and UC even in the absence of infection, but I could not find any mention of CKD. Please review m again.

Ref 24 reports that PCT levels have been found to be falsely elevated in patients suffering from various degrees of chronic kidney disease.

Author Response

The authors retrospectively examined the factors affecting AKI recovery in single centers in Mexico. At first, you described higher Cr level, lower eGFR, maximum serum Cr, higher FENa

24 UP, higher potassium, and proclcitonin (PCT) showed significant in univariate analysis.

But in multivariate analysis, only minimum eGFR (maximum Cr?) showed independent risk factor for non-remitting AKI.

So all of the other factors (higher Cr level, lower eGFR, maximum serum Cr, higher FENa

24 UP, higher potassium, and proclcitonin) may be confounding, which leads to slightly different conclusions.

Show the details of the multivariate analysis in a table. How many variables did you use in multivariate analysis?  Since serum Cr and eGFR are almost synonymous with each other, it is better to use only one of them as a variable to reduce the number of variable in multivariate analysis.

We appreciate the suggestion regarding multivariate analysis. Following your suggestion, for the multivariate analysis, we worked only with seven variables, eliminating the eGFR, eGFR min, AKI 3, and FENa < 1% since they presented multicollinearity with the rest of the variables used, making it impossible to obtain a reliable result. Said variables presented a variance inflation factor (VIF) of 10 or close to 10, verifying the linear correlation. We present the results of the analysis in a new table (Table 4), as well as make a clearer interpretation in the discussion. We also adjusted the text in methods and results to reflect this.

Data that do not show a normal distribution have been corrected from average to median. Let's also modify (minimum-max) to (IQR).

The results where medians are reported were modified to use the IQR instead of the min-max. 

FENA→FENa  (in Discussion   L194)

FENA was replaced into FENa in the text.

In Discussion

Ref 25 mentioned that Procalcitonin is elevated in inflammatory bowel disease such as CD and UC even in the absence of infection, but I could not find any mention of CKD. Please review m again.

Ref 24 reports that PCT levels have been found to be falsely elevated in patients suffering from various degrees of chronic kidney disease.

Indeed, reference number 25 was used to justify that PCT levels are elevated in inflammatory diseases even in the absence of active infection, and that specific reference does not mention CKD. The reference is replaced with a more appropriate one for the context of the text, which supports the finding of elevated PCT in addition to adding a possible pathophysiological explanation.